# The realization logic of rural revitalization: Coupled coordination analysis of development and governance

Hongxun Xiang[1]*, Boleng Zhai[1], Yang Yang[2]

**1** School of Public Administration, Sichuan University, Chengdu, Sichuan, China, **2** School of Marxism, Xichang Minzu Preschool Normal College, Xichang, Sichuan, China

* xianghx@stu.scu.edu.cn

**Data Availability Statement:** All relevant data are within the manuscript and its Supporting information files.

**Funding:** The author(s) received no specific funding for this work.

## Abstract

### Background

Socialism with Chinese characteristics has entered a new stage. The principal social contradiction is the uneven development of urban and rural areas. The rural revitalization strategy has emerged as time has required. The realization of rural revitalization not only requires development to lay the foundation of the countryside but also requires governance to lead the development of the countryside. Development and governance are two indispensable aspects of rural revitalization. However, China's rural areas have long been in a state of development without governance, and this situation must change. Therefore, systematically exploring the relationship between development and governance is the key to solving the current shortcomings in rural areas.

### Methods

Based on the data from the statistical yearbook, the study constructed a set of evaluation indicators for rural development governance and revitalization and verified the model's effectiveness.The entropy method and the assessment model were used to calculate the comprehensive score of rural development, governance, and revitalization. The relationship between rural development and governance was analyzed using a coupled coordination model. The regression analysis model was used to explore the relationship between the coupling results of rural development, governance, and rural revitalization.

### Results

From the comprehensive results, both development and governance show an upward trend, but the upward trend of development is better. From the analysis of coupling coordination between development and governance, the C value is in good condition, the T value fluctuates wildly, and the D fluctuates with the fluctuation of T. Judging from the comprehensive score of rural revitalization, it also shows an upward trend year by year. Judging from the regression analysis results of coupling coordination degree and rural revitalization

**Competing interests:** The authors have declared that no competing interests exist.

comprehensive score, coupling coordination degree will significantly impact the rural revitalization evaluation value.

## Conclusions

The study found that current rural development and governance present a spiral coupling coordination relationship, and the degree of coupling coordination significantly correlates with rural revitalization. Based on the research conclusions, the study further proposes three paths to promote the coupling and coordination of development and governance. The first is an organizational isomorphism, which builds a coupled coordination system for rural development and governance. The second is to tilt resources and improve the supply of connected and coordinated factors for rural development and governance. The third is the operating mechanism to optimize rural development and governance's coupling and coordination path.

## Introduction

### Background

High-quality development and high-efficiency governance are new propositions, requirements, and missions based on the modernization of grassroots governance with Chinese characteristics in the new era. They are the key strategy for breaking China's urban-rural dual pattern and achieving integrated urban-rural development [1]. China's rural economic development and social governance mainly rely on grassroots party organizations, village committees, and collective economic organizations [2]. Judging from the division of labor among the three, the main goal of grassroots party organizations is to strengthen the identity of grassroots party organizations, the identity of the macro-state, and the overall leadership of rural areas, agriculture, and farmers' "three rural" work [3]. As a grassroots autonomous organization, the core function of the village committee is to mobilize villagers' participation and promote comprehensive rural construction, development, services, governance, and other matters [4]. The core task of collective economic organizations is to develop and expand the village's collaborative economy while enriching the financial income of villagers [5]. However, judging from the development process of China's rural areas, China's rural areas have long been in a situation of development but no governance. For a long time, rural China has focused mainly on economic development [6]. Economic development has gone through the road of collective economy, the household responsibility system, the cooperative model, the route of agricultural industrialization, and the collaborative economy has been developed and strengthened [7]. Compared with economic development, although the proposal and implementation of the rural revitalization strategy and the grassroots governance modernization strategy are gradually being paid attention to, rural governance issues are progressively being paid attention to. However, development and governance are affected by problems such as diverse organizations, fragmentation, and dispersed resources, making it difficult to carry out systematic planning and forward-looking layouts for rural development and governance. Rural development and governance are in a state of fragmentation [8].

Specifically, from a development perspective, by the end of 2022, 55% of the state's rural subsidy funds will be used directly for rural development. From a development perspective, 55% of the state's rural subsidy funds are directly used for rural development. Judging from

development results, the grain harvest loss rate is controlled within 3%. Livestock and poultry manure, straw, and agricultural film utilization rates exceed 78%, 88%, and 80%, respectively. The domestic waste recycling and processing rate reaches 90%, and the nearby employment rate of villagers exceeds 90% [9]. Understandably, rural construction and development have achieved gratifying results. However, in the agricultural industrialization development model with cooperatives as the organizational model, its business entities focus on rural economic growth and increasing rural income. They cannot embed institutional norms, management technologies, and internal and external resources in industrial development into rural governance [10].

From the governance perspective, with the implementation of grassroots governance modernization and rural revitalization, attention to rural governance continues to rise. Integrating departmental resources such as political and legal resources, civil affairs, and organizational systems reforms the rural governance system. It enhances rural governance capabilities through information technology, resource allocation, and talent supply [11]. Regarding governance effects, the level of public services such as rural education, medical and health care, and elderly care services has been qualitatively improved, and governance methods such as the point system, the list system, digitization, and intelligence have been comprehensively promoted [12]. However, rural governance is more like an independent sector, lacking linkage with agriculture and rural areas, collective organizations, economic entities, etc. That is to say, the efficiency of rural revitalization is improved through the coupling of the organization, system, structure, power, and capabilities of the development and governance system [13].

The No. 1 Central Document in 2023 pointed out that developing modern rural service industries, cultivating new rural industry formats, and strengthening county-level initiatives to enrich people will promote the high-quality development of rural industries through village planning, living environment improvement, infrastructure construction, and service capacity improvement. Promote the construction of a livable, industrial, and beautiful countryside, and improve the organizational leadership system to improve the efficiency of rural governance [14]. High-quality rural development and high-efficiency governance have become the core proposition of future rural development. They are the fundamental guarantee for achieving China's economic and social development goals during the "14th Five-Year Plan" period. They are the key to solving the shortcomings and weaknesses of rural revitalization. Based on this, realizing the rural revitalization strategy as scheduled urgently requires development and governance. Exploring the coupling and coordination relationship between rural development and governance and the connection between the coupling coordination relationship and rural revitalization are the keys to solving the above problems. The answers to this series of questions are significant. In theory, the economic development theory and social governance theory are integrated and introduced into the rural field, laying a foundation for the expansion of the theory. In practice, clarifying the relationship between development and governance and proposing targeted measures to promote the coupling of rural development and governance are the keys to realizing rural revitalization as scheduled.

## Literature review

The academic community has carried out many studies on rural development. Some scholars explore the fields of rural development, from rural tourism [15], collective economy [16], professional cooperatives [17], agricultural industrialization [18], urban-rural integration [19], etc. Studying the endogenous power issues of rural development from the perspective of other actor strategies [20], subject endogenous power [21], and urban spillover effects [22], it is pointed out that the endogenous power of rural development comes from the mobilization of

subjects to gather action forces, industrial integration to reshape the form of kinetic energy, and the rational deepening of value recognition by village communities [23]. Find factors affecting rural development, from rural infrastructure, government capabilities, industrial structure, labor quality, etc [24]. They constructed rural ecological environment measurement indicators, rural development assessment indicators, rural dynamic assessment indicators, etc [25]. Scholars have conducted many studies on rural development from different dimensions, laying a theoretical foundation for research. In the Chinese field, scholars have systematically discussed the relationship between rural development and shared prosperity [26], believing that the key to realizing shared prosperity lies in the countryside [27] and the key to realizing rural revitalization lies in narrowing the gap between urban and rural areas [28]. In addition, some scholars have systematically summarized the experience of foreign rural development [29]. However, the current academic research on rural development only stays in the development field. It selectively ignores the significant impact of rural governance on development, which needs to be further expanded.

Compared with research on development, there are relatively few studies on rural governance, which mainly focus on four aspects. The first is the study of rural governance theory. Holistic governance theory believes that organizational fragmentation occurs due to the lack of effective communication, negotiation, and cooperation between organizations [30], as well as the asymmetry of information between organizations and the maximization of self-interest [31]. This inevitably leads to integration difficulties in rural areas and affects the efficiency of rural governance [32]. Collaborative governance theory believes collaboration is between systems, a process from disorder to order [33]. Due to the large number of governance entities in rural areas and the overemphasis on their interests [34], the performance of rural social governance is directly affected [35].

Second, the content dimension of rural governance. From the perspective of party building leading rural governance, scholars explore the two-way empowerment logic of party building and rural governance [36], the path for transforming political potential energy into development efficiency [37], and how party building leads the construction of rural communities [38]. Discuss community governance models, community governance content, new collectivism, etc., from the perspective of rural governance innovation [39]. Discuss the connotation, scope, construction reasons, paths, and advantages of community construction from the standpoint of community construction [40]. Discuss the role of technology integration, problems of technology integration, dilemmas of technology integration, and the advantages of technology integration from the perspective of technology governance [41].

Third, research on the rural governance system. Based on the governance pattern of co-construction, co-governance, and sharing, scholars propose that the key to the "integration of three governance" is autonomy as the basis [42], the rule of law as the basis, and the rule of morality as the first. The key to the rural governance system is based on market logic, government logic, and multiple Competitive logic, building a new rural governance system, focusing on cadre teams' construction.

Finally, there is research on rural governance capabilities. Scholars elaborate on rural governance capabilities from two dimensions: specific connotation and practical form [43]. It also innovates the dimensions of governance capabilities and regards social management, system integration, and policy execution capabilities as the main factors affecting governance capabilities [44]. It is pointed out that the key to improving rural governance capabilities is improving overall planning and mass work capabilities [45]. At the same time, it is believed that under the realistic background of comprehensive rural revitalization, the weakening leadership of grassroots party organizations, insufficient integration of grassroots party building, and unreasonable structure of grassroots cadres are the most critical limiting factors [46]. From this, it

can be found that the research on rural governance is also separated and developed, mainly focusing on specific dimensions such as governance theory, content, system, and capabilities.

When sorting out the research context of rural development and governance, it is easy to find that 2017 is the dividing point for rural development and governance research. Previous research focused on rural development, but later on, both development and governance gradually emerged, and the research perspective also changed from development to development plus governance. In terms of research content, there is both development and governance. In terms of research scope, there is more emphasis on cross-differentiation. In terms of research methods, it is more biased towards the combination of norms and empirical studies, and in terms of research level, it is more in-depth. However, there are still weak links in research on development and governance, which require further research. Scholars focus on either rural development or rural governance. However, from the macro level of rural revitalization, very few studies comprehensively consider the embedded relationship between rural development and governance. However, realizing rural revival is a comprehensive process that requires both economic progress brought by development and efficiency improvement brought by governance. Therefore, the urgent need for practice brings about the possibility of theoretical exploration. The study is based on the realistic needs of coupling rural development and governance. It uses mathematical models to explore the coupling and coordination relationship between rural development and rural governance. In addition, the coupling coordination relationship and rural revitalization results are combined to fit, and the intrinsic relationship between the coupling results of development and governance and rural revitalization is explored. Based on the research results, answer the evolution trend of coupled coordination of rural development and governance, the relationship between related coordination of development and governance and rural revitalization, and the optimization path of "development + governance" coupled coordination to achieve rural revival.

## Materials and methods

### Research framework

Based on the research hypothesis, two problems need to be solved. One is the evolutionary trend of the coupling coordination relationship between rural development and governance, and the other is the relationship between coupling coordination degree and rural revitalization. It should be noted that the measurement of rural development and governance is too large. Therefore, rural development and governance measurement focuses on the resource dimension. That is to explore the coupling and coordination relationship between rural development and governance from the dimensions of rural development and governance resources. First, the research results on rural development resources and governance resources based on the existing literature will be studied, the principle of data availability will be followed, and an evaluation index system for rural development resources and governance resources will be constructed. Through the input of panel data, we use the entropy method to calculate rural development resources and the assessed value of governance resources. Secondly, based on the evaluation values of rural development resources and governance resources, combined with the coupling coordination analysis model, the coupling situation of rural development resources and governance resources was calculated year by year, and the coupling development trend of rural development resources and governance resources was obtained. Thirdly, combined with the country's analysis of rural revitalization, rural revitalization is divided into five major sectors: organizational revitalization, industrial revitalization, talent revitalization, ecological revitalization, and cultural revitalization. Based on this, a rural revitalization evaluation model is constructed to obtain year-by-year rural revitalization results. Finally, based on

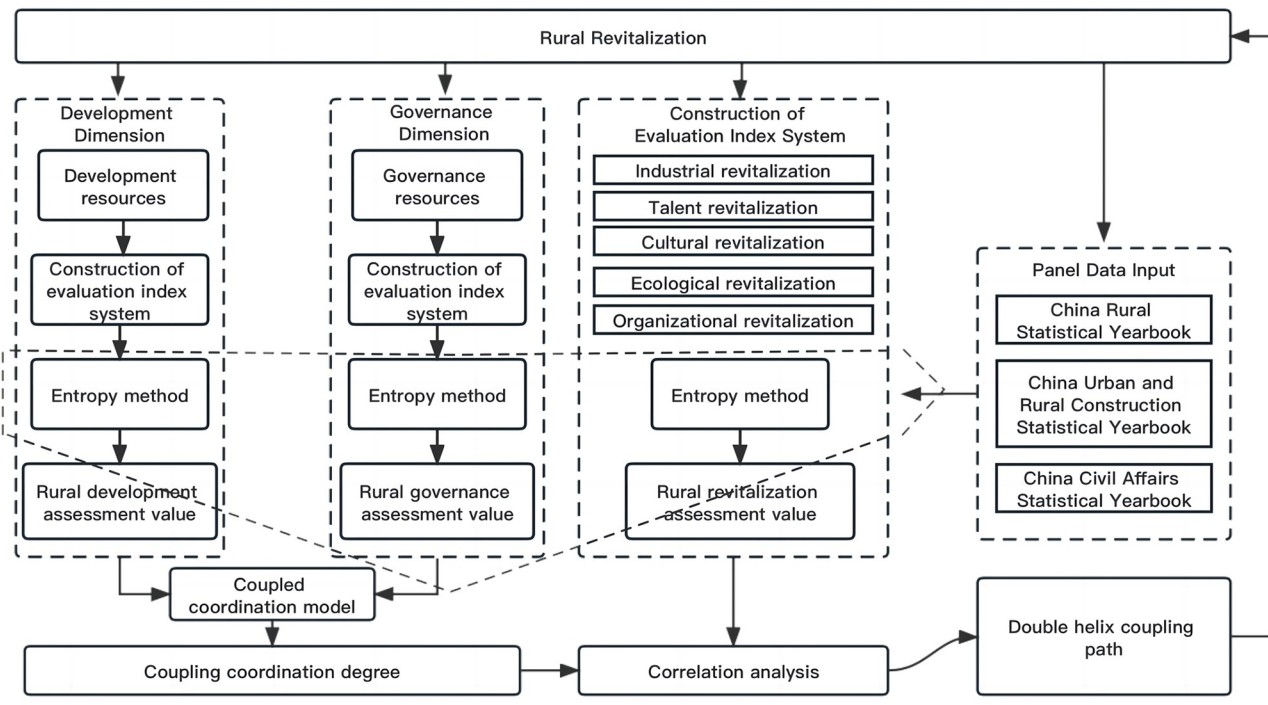

**Fig 1. Research framework design.**

the coupling coordination analysis results of rural development resources and governance resources combined with the rural revitalization assessment, the correlation between the two was analyzed using the correlation degree. At the same time, considering the availability and scientific nature of data, the study finally used official statistics (Fig 1).

## Evaluation indicators

The development of resources not only involves economic growth but is also a comprehensive process, including economic development resources, and involves various aspects such as society, culture, and ecology. However, among the existing research results, there are few measurements of rural development resources, and more are measured around the dimensions of rural revitalization, industrial development, digital countryside, ecological countryside, and urban-rural integration [47]. Regarding the measurement dimensions, existing research results focus on economic development, industrial cooperation, cultural development, environmental development, spiritual civilization, etc [48]. However, the existing dimensions only focus on a particular aspect of development resources, such as culture, ecology, industry, etc., which is inconsistent with the purpose of the research. The objective data required for research on factors such as politics, economy, culture, society, ecology, and spiritual civilization that measure rural development cannot be obtained from statistical yearbooks, and the feasibility could be higher. Therefore, this study treats rural development resources and governance resources as a pair of corresponding concepts that distinguish them, and the indicators involved are all observable economic indicators. In addition, to meet the needs of multiple assessment models, the data range and units of the rural development resources and governance resource indicators involved in the study were transformed into consistent ones. Individual indicators are subject to certain modifications based on accessibility concerning relevant policy documents.

Based on the statistical yearbook data, the study screened specific measurement indicators, compared the obtained indicators to obtain secondary indicators, and then classified them to get first-level indicators. The classification of first-level indicators is mainly based on resource characteristics and system theory. From the perspective of resource characteristics, there are not only funds, talents, village-specific resources, etc., but also financial funds, social capital, self-raised funds, etc. From the systems theory perspective, the life cycle of resources includes processes such as input, transformation, and output. Based on this, the study divided development and governance resources into three categories: Input type, endogenous type, and output type, based on the above characteristics, as shown in Table 1 below.

Compared with the inconsistency in the measurement indicators of rural development and rural governance, the academic community has formed a relatively unified understanding of the measurement dimensions of rural revitalization, which benefits from the clarity of policy documents. In the report of the 19th National Congress of the Communist Party of China in 2017, it was clearly stated that the future blueprint for rural revitalization is prosperous industries, livable ecology, civilized rural customs, effective governance, and affluent life. In 2018, when General Secretary Xi Jinping participated in the deliberation of the Shandong delegation during the National Two Sessions, he once again reiterated the five cores of implementing the

**Table 1. Rural development resources and governance resource indicator system.**

| System | First level indicator | Secondary indicators | Data Sources |
|---|---|---|---|
| Development Resources | Input type resource | Agricultural funds | China Rural Statistical Yearbook |
| | | Water conservancy funds | China Rural Statistical Yearbook |
| | | Comprehensive reform funds | China Rural Statistical Yearbook |
| | | Fixed assets investment | China Rural Statistical Yearbook |
| | Output type resource | Road blackening length | China Urban and Rural Construction Statistical Yearbook |
| | | Ditch regulation length | China Urban and Rural Construction Statistical Yearbook |
| | | Contribution rate to scientific and technological progress | China Rural Statistical Yearbook |
| | Endogenous type resource | Village permanent population | China Urban and Rural Construction Statistical Yearbook |
| | | Villager literacy (high school and above) | China Rural Statistical Yearbook |
| | | Number of cooperatives | China Rural Statistical Yearbook |
| | | Cultivated area | China Rural Statistical Yearbook |
| Governance Resources | Input type resource | Investment in municipal public facilities | China Urban and Rural Construction Statistical Yearbook |
| | | Comprehensive service facility coverage | China Rural Statistical Yearbook |
| | | Full-time community worker | China Urban and Rural Construction Statistical Yearbook |
| | Output type resource | Collective business income | China Rural Statistical Yearbook |
| | | Public building area | China Urban and Rural Construction Statistical Yearbook |
| | | Comprehensive service organization | China Civil Affairs Statistics Yearbook |
| | Endogenous type resource | Education level of village cadres (junior college) | China Civil Affairs Statistics Yearbook |
| | | Proportion of number of village cadres and party members | China Civil Affairs Statistics Yearbook |
| | | Rule of law construction | China Civil Affairs Statistics Yearbook |
| | | Employees in the village | China Civil Affairs Statistics Yearbook |
| | | Villagers' willingness to participate | China Civil Affairs Statistics Yearbook |

**Table 2. Rural revitalization indicator system.**

| System | First level indicator | Secondary indicators | Data Sources |
|---|---|---|---|
| Rural Revitalization | Industrial revitalization | Agricultural machinery total power | China Rural Statistical Yearbook |
| | | Gross output value of agriculture, forestry, animal husbandry and fishery | China Rural Statistical Yearbook |
| | | Agricultural Producer Price Index | China Rural Statistical Yearbook |
| | Talent revitalization | Number of rural employees | China Civil Affairs Statistics Yearbook |
| | | Cultural literacy (high school and above) | China Rural Statistical Yearbook |
| | | Rural and township health personnel | China Rural Statistical Yearbook |
| | Cultural revitalization | Coverage rate of comprehensive service facilities in rural communities | China Civil Affairs Statistics Yearbook |
| | | Number of full-time teachers in ordinary high schools | China Rural Statistical Yearbook |
| | | Number of students enrolled in ordinary high schools | China Rural Statistical Yearbook |
| | Ecological revitalization | Crop damage area | China Rural Statistical Yearbook |
| | | Ditch regulation length | China Urban and Rural Construction Statistical Yearbook |
| | | Contribution rate to scientific and technological progress | China Rural Statistical Yearbook |
| | Organizational revitalization | Investment in comprehensive rural reform | China Rural Statistical Yearbook |
| | | Comprehensive community service agencies and facilities | China Civil Affairs Statistics Yearbook |
| | | Villagers self-government organization | China Civil Affairs Statistics Yearbook |

rural revitalization strategy, namely industrial revitalization, talent revitalization, cultural revitalization, ecological revitalization, and organizational revitalization. This provides clear guidance for the academic community to evaluate the current results of rural revival. The study formed an evaluation index system based on the five core strategies of rural revitalization and the principle of data accessibility. That is to say. The study adopts the "from both ends to the middle" strategy. First, based on national standards, the target dimension of rural revitalization is divided into five primary indicators: industry, talent, culture, ecology, and organization. Secondly, the evaluation indicator set is extracted based on the data from China Rural Statistical Yearbook, China Urban and Rural Statistical Yearbook, and China Civil Affairs Statistical Yearbook. Finally, according to the indicator classification standards, the extracted indicator sets are classified into first-level indicators to form second-level evaluation indicators, thus constructing a rural revitalization evaluation indicator system, as shown in Table 2 below.

## Research methods

Before model measurement, the indicators need to be processed due to differences in indicator dimensions and units. Combined with the design of the research framework, the indicators need to reflect the changing trend of individuals as a whole. Therefore, the study adopts the summation and normalization processing method to perform dimensionless processing on the extracted data (1). To solve the research hypothesis problem. First, need to measure the coupling and coordination relationship between rural development and governance, but the development and governance dimensions have different elements. Therefore, the study uses the entropy method to measure the two dimensions of development and governance to form the overall results of development and governance in the ten years from 2012 to 2021. It is necessary to measure the relationship between development and governance. Still, development and governance are both dynamic dimensions, so the coupling coordination model is used to measure and determine the coupling coordination relationship between the two. Secondly, the perspective of rural revitalization assessment is consistent with that of rural development and

governance, which is also conducted using the entropy method. Finally, compared with the coupling coordination value of rural development and governance, the rural revitalization assessment result is a set of established comparative data. That is, the degree of coupling coordination is the independent variable, and the results of the rural revitalization assessment are the dependent variables. Therefore, this study uses a regression analysis model to explore the relationship between coupling coordination outcomes and rural revitalization. In summary, research requires the comprehensive use of the entropy method, coupled coordination model, and regression analysis model.

$$SN = \frac{X}{\sum_{i=1}^{n} X_i}, \tag{1}$$

**Entropy method.** Entropy is a measure of uncertainty. Generally speaking, the greater the information, the smaller the delay and the smaller the entropy value, and vice versa [49]. Therefore, the entropy method uses the amount of information carried to calculate the weight and then calculates the entropy score to provide a basis for the comprehensive evaluation of indicators [50]. Generally speaking, the data needs to be forward or inverse-normalized before performing entropy analysis. However, the research has processed and met the data [0–1] distribution requirements, so the entropy value formula is directly used for calculation. Assume that "m" indicators and "n" samples are finally selected, and Xij represents the "j-th" indicator of the "i-th" sample (i = 1,2,3...,n; j = 1,2,3...,m), "P" means the sample weight, "e" means the information entropy value, "d" represents the difference coefficient, "W" represents the index weight, and "Z" represents the comprehensive score. The calculation formula is as shown in (2) (3) (4) (5) (6) (7).

$$P_{ij} = \frac{X_{ij}}{\sum_{i=1}^{n} X_{ij}}, \tag{2}$$

$$e_j = -K * \sum_{i=1}^{n} (P_{ij} * \ln P_{ij}), \tag{3}$$

$$K = \frac{1}{\ln n}, \tag{4}$$

$$d_j = 1 - e_j, \tag{5}$$

$$W_j = \frac{d_j}{\sum_{j=1}^{m} d_j}, \tag{6}$$

$$Z_j = \sum_{j=1}^{m} W_j X_{ij}, \tag{7}$$

**Coupling coordination model.** The coupling coordination degree mainly measures three indicators: coupling degree C, coordination index T, and coupling coordination degree D [51]. Among them, the degree of coupling refers to the interaction between two or more systems and the dynamic correlation that achieves coordinated development. It reflects the degree of interdependence and mutual constraints between systems. The degree of coordination refers to the degree of benign coupling in coupling interactions and is used to indicate the quality of coordination [52]. The value range of the coupling coordination degree is [0, 1]. The

larger the value, the better the coupling coordination degree, and vice versa. The calculation formula is as shown in (8) (9) (10).

$$C = 2 * \left( \frac{U_1 * U_2}{U_1 + U_2} \right)^{\frac{1}{2}}, \tag{8}$$

$$T = a_1 U_1 + a_2 U_2, \tag{9}$$

$$D = \sqrt{C * T}, \tag{10}$$

Among them, a1 and a2 represent the weight of rural development and governance resources, respectively, a1+a2 = 1. Based on existing policy documents and research results, the study believes that rural development and governance are equally important, so both values are 0.5 [53]. In addition, the study refers to the classification standard of coupled coordination model levels. The coupling and coordination degree of rural development and governance is divided into ten classes, as shown in Table 3.

**Regression analysis model.** Regression analysis is mainly used to study the relationship between quantitative data, a statistical analysis method that determines the interdependent quantitative relationship between two or more variables [54]. Regression analysis has different classification standards. According to the number of variables involved, it can be divided into single regression and multiple regression [55]. According to the number of dependent variables, it can be divided into simple regression and multiple regression [56]. The relationship between independent and dependent variables can be divided into linear and nonlinear regression [54]. According to the definition of the research hypothesis, this article studies the relationship between a single independent variable and a single dependent variable, and the data is linearly distributed. Therefore, the regression analysis model involved in the study is a linear regression model. Suppose x is the independent variable and y is the dependent variable. The linear regression equation is calculated as follows (11) (12) (13).

$$\hat{b} = \frac{\sum_{i=1}^{n}(x_i - \bar{x})(y_i - \bar{y})}{\sum_{i=1}^{n}(x_i - \bar{x})^2} = \frac{\sum_{i=1}^{n}x_i y_i - n\bar{x}\bar{y}}{\sum_{i=1}^{n}x_i^2 - n\bar{x}^2}, \tag{11}$$

**Table 3. Criteria for classification of coupling coordination levels.**

| Coupling coordination degree D value interval | Coordination level | Degree of coupling coordination |
|---|---|---|
| $(0.0 \sim 0.1)$ | 1 | Extremely disordered |
| $[0.1 \sim 0.2)$ | 2 | Severe disorder |
| $[0.2 \sim 0.3)$ | 3 | Moderate disorder |
| $[0.3 \sim 0.4)$ | 4 | Mild disorder |
| $[0.4 \sim 0.5)$ | 5 | On the verge of disorder |
| $[0.5 \sim 0.6)$ | 6 | Barely coordinated |
| $[0.6 \sim 0.7)$ | 7 | Junior coordinator |
| $[0.7 \sim 0.8)$ | 8 | Intermediate level coordination |
| $[0.8 \sim 0.9)$ | 9 | Well coordinated |
| $[0.9 \sim 1.0)$ | 10 | High quality coordination |

$$\hat{a} = \bar{y} - \hat{b}\bar{x}, \tag{12}$$

$$\hat{y} = \hat{b}x + \hat{a}, \tag{13}$$

## Data sources

Since 2012, the country has successively promulgated the "Opinions on Strengthening Urban and Rural Community Governance," "Rural Revitalization Strategic Plan 2018–2022", "Rural Construction Action Implementation Plan," "Opinions on Strengthening the Modernization of Grassroots Governance Systems and Governance Capacities" and the central government's policies over the years—document No. 1, etc. Under the guidance of policies, rural development, and governance have ushered in significant opportunities, especially rural resources, organizations, and human resources, which have been continuously strengthened. At this stage, the policy concept focuses on high-quality development and satisfying the people's pursuit of a better life, the policy path emphasizes the integration of urban and rural areas, and the policy goals focus on rural revitalization and grassroots governance modernization. The starting point of all this was the successful convening of the 18th National Congress of the Communist Party of China, that is, after 2012. To this end, the policy evolution nodes of rural development and governance and the availability of relevant data are comprehensively considered. The study takes 2012–2021 as the research period. It uses the China Rural Statistical Yearbook, China Urban and Rural Statistical Yearbook, and China Civil Affairs Statistical Yearbook as data sources to research the relationship between China's rural development and resource coupling coordination effects, coupling coordination, and rural revitalization effects.

## Results

### Rural development and governance comprehensive score

Through the calculation of formula (1) (2) (3) (4) (5) (6) (7), the comprehensive score of rural development and governance is shown in Table 4. The following conclusions can be drawn from the data in the table: First, on the whole, whether it is the development dimension or the governance dimension, the overall evolution trend is upward. Second, specifically, the evolution trend of development is better than the evolution trend of governance. Third, looking at the stages, the comprehensive governance score rose rapidly from 2012 to 2014 but fluctuated wildly from 2014 to 2020. This stage is when the country implements the targeted poverty alleviation strategy. This stage is when the government implements the targeted poverty alleviation strategy. The country's primary energy is to invest in development resources, develop rural industries, and strive to realize a moderately prosperous society. After 2020, the comprehensive score of governance has increased significantly, even exceeding the comprehensive score of development. This is closely related to the completion of the construction of a vast, moderately prosperous society and the implementation of the rural revitalization strategy. The country has begun to focus on investment in rural organization construction, talent introduction, villagers' autonomy, and other governance dimensions. Unlike the fluctuation trend of

**Table 4. Comprehensive score of rural development and governance.**

|  | 2012 | 2013 | 2014 | 2015 | 2016 | 2017 | 2018 | 2019 | 2020 | 2021 |
|---|---|---|---|---|---|---|---|---|---|---|
| Development resources | 0.054 | 0.062 | 0.069 | 0.079 | 0.086 | 0.112 | 0.120 | 0.130 | 0.140 | 0.149 |
| Governance resources | 0.062 | 0.085 | 0.122 | 0.084 | 0.067 | 0.095 | 0.110 | 0.089 | 0.095 | 0.191 |

the governance comprehensive score, the development comprehensive score has steadily increased since 2012. This is consistent with China's development logic. China has always insisted that development is the last word.

## Coupling and coordination degree of rural development and governance

Based on the results of the comprehensive score of rural development and governance, through formulas (8) (9) (10), the coupled and coordinated development of the comprehensive score of rural development and governance from 2012 to 2021 is calculated as shown in Table 5. As can be seen from the table, the coupling degree C value is in good condition, basically fluctuating around 0.9, and the difference between years is negligible. Compared with the coupling degree C, the coordination degree T value fluctuates wildly, and the average value is lower. Except for 2021, the other years are lower than 0.6. From the perspective of the coupling coordination degree D, the coordination index T has an essential impact on it, causing D to fluctuate with the fluctuation of T. Judging from the results, this may be closely related to the convening of the 18th National Congress of the Communist Party of China in 2012 and the start of the 13th Five-Year Plan. Since the 18th National Congress, the response speed of rural development and governance coupling has been breakneck. Still, it may be affected by the limitations of the rural environment. The coordination degree or stability of development and governance could be better, which is also why the coupling coordination degree value is low. With the development of rural revitalization, the country attaches great importance to the investment of rural development resources and the investment of rural governance resources. Therefore, after 2020, the coordination index T of rural development and governance resources soared to 0.99, and the coupling coordination value also reached 0.995, which is in the high-quality coordination stage. However, this may be due to policy dividends, and the degree of coupling coordination in the later period may continue to show a fluctuating development trend of spiral coupling.

## Comprehensive score of rural revitalization assessment

Through the calculation of formula (1) (2) (3) (4) (5) (6) (7), the comprehensive score result of the rural revitalization assessment is shown in Fig 2 below. It can be concluded from the figure that the rural revitalization assessment results have shown an increasing trend year by year since 2012. However, there were apparent changes in 2018, and its growth rate suddenly became faster. Before 2018, the country had yet to formally propose a formal rural revitalization strategy. China's rural areas were undergoing targeted poverty alleviation work at that time. The favorable external effects of targeted poverty alleviation have caused the evaluation

**Table 5. Calculation results of coupling coordination degree.**

| Year | Coupling degree C value | Coordination index T value | Coupling coordination degree D value | Coordination level | Degree of coupling coordination |
|------|------------------------|----------------------------|--------------------------------------|--------------------|--------------------------------|
| 2012 | 1.000 | 0.010 | 0.100 | 2 | Severe disorder |
| 2013 | 0.934 | 0.136 | 0.356 | 4 | Mild disorder |
| 2014 | 0.877 | 0.314 | 0.525 | 6 | Barely coordinated |
| 2015 | 0.982 | 0.219 | 0.463 | 5 | On the verge of disorder |
| 2016 | 0.687 | 0.193 | 0.365 | 4 | Mild disorder |
| 2017 | 0.919 | 0.431 | 0.629 | 7 | Junior coordinator |
| 2018 | 0.956 | 0.531 | 0.712 | 8 | Intermediate level coordination |
| 2019 | 0.825 | 0.502 | 0.643 | 7 | Junior coordinator |
| 2020 | 0.837 | 0.580 | 0.697 | 7 | Junior coordinator |
| 2021 | 1.000 | 0.990 | 0.995 | 10 | High quality coordination |

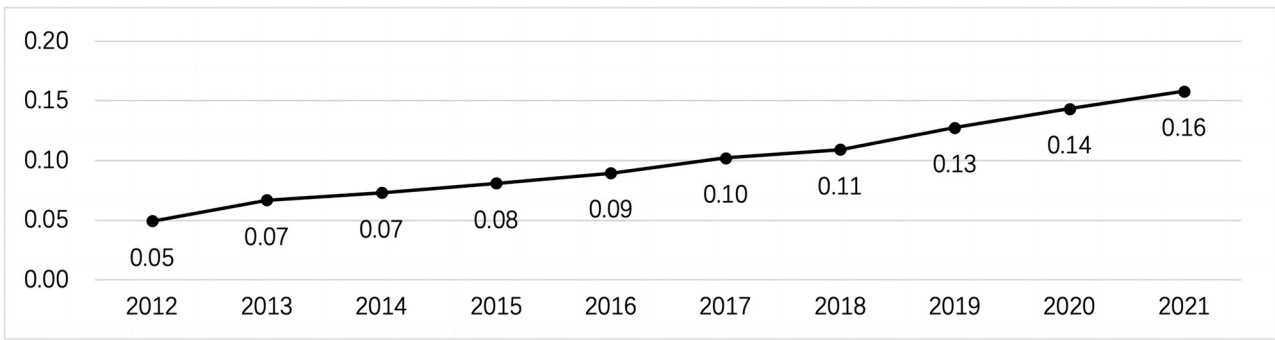

**Fig 2. Comprehensive score trend of rural revitalization assessment.**

results to increase yearly. In 2018, the country officially implemented the rural revitalization strategy, and rural revitalization became formally the core focus of rural work. Based on this, after 2018, policies, resources, talents, etc., were more concentrated and achieved better results, which suddenly caused the growth rate of evaluation results to increase.

## Regression analysis of coupling coordination degree and rural revitalization comprehensive score

The study combines the evaluation value of the coupling coordination degree of rural governance and rural development with the evaluation value data of rural revitalization. It uses formulas (11) (12) (13) to calculate the regression relationship between the two. The study conducted a correlation analysis on the rural revitalization evaluation value and the coupling coordination degree and determined a correlation between the two, and the scatter distribution is linear. Therefore, the research hypothesis requirements are met, and the two show a linear correlation, which can be used for linear regression analysis. Combined with the research hypothesis, the dependent variable of linear regression analysis is the comprehensive assessment value of rural revitalization, and the independent variable is the coupling coordination degree of rural development and governance. The calculation results are in the table below (Table 6). As can be seen from the table below, the model formula can be written as rural revitalization assessment value = 0.029 + 0.129 * coupling coordination degree. The R-squared value of the model is 0.821, which means that the coupling coordination degree can explain 82.1% of the variation in the rural revitalization assessment value. This shows that coupling rural development and governance is crucial to the scheduled realization of rural revitalization. When the F test was performed on the model, it was found that it passed the F test (F = 36.780,

**Table 6. Linear regression analysis results.**

|  | Unstandardized coefficient | | Standardized coefficient | t | p | Collinearity Diagnosis | |
|---|---|---|---|---|---|---|---|
|  | B | Standard error | Beta |  |  | VIF | Tolerance |
| Constant | 0.029 | 0.013 | - | 2.318 | 0.049* | - | - |
| Coupling coordination degree | 0.129 | 0.021 | 0.906 | 6.065 | 0.000** | 1.000 | 1.000 |
| R2 | 0.821 |  |  |  |  |  |  |
| Adjust R2 | 0.799 |  |  |  |  |  |  |
| F | F (1,8) = 36.780,p = 0.000 |  |  |  |  |  |  |
| D-W | 1.545 |  |  |  |  |  |  |

* p<0.05 ** p<0.01

p = 0.000<0.05), which means that the coupling coordination degree will impact the rural revitalization evaluation value. The final detailed analysis shows that the regression coefficient value of the coupling coordination degree is 0.129 (t = 6.065, p = 0.000<0.01), which means that the coupling coordination degree will significantly positively impact the rural revitalization evaluation value. It can be concluded that the coupling of rural development and governance will significantly impact the realization of rural revitalization.

## Discussion

Since modern times, China has been facing the problem of poverty for a long time. Therefore, since the founding of New China, development issues have been emphasized, and this is also true for rural areas. However, compared with development, governance has yet to receive enough attention, and development and governance have been out of balance for a long time. The structural deviation caused by this problem has seriously affected the process of rural revitalization. Although the developed East has explored rural governance, it has provided a reference for rural governance in underdeveloped areas. Ultimately, it is still necessary to systematically answer the question of the relationship between development and governance.

### Rural development and governance present a spiral coupling and coordination relationship

Based on the coupling coordination degree calculation results in Table 5, study and draw the following Fig 3. The graph uses the X-axis as the axis of symmetry and is symmetrical up and down. The lines represent the coupling degree, coordination degree, and coupling

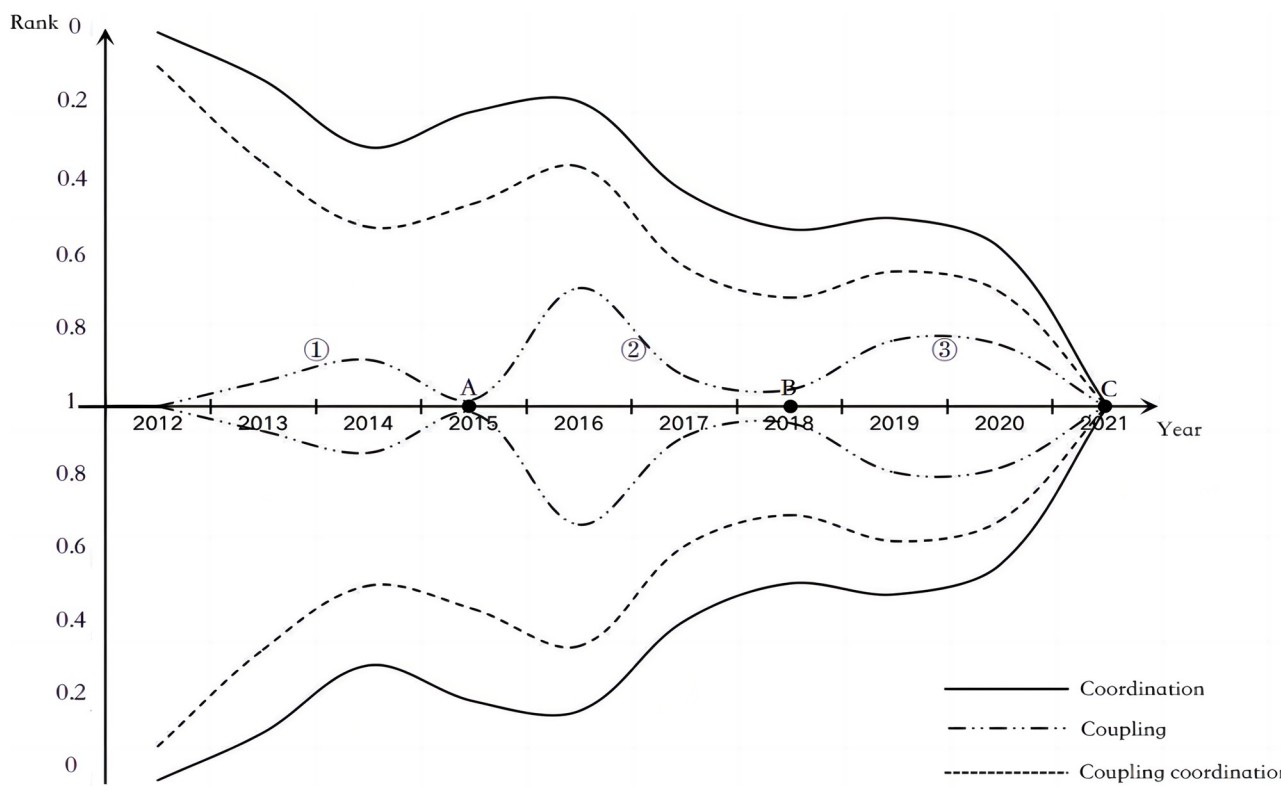

**Fig 3. Trend chart of coupling and coordination degree of rural development and governance.**

coordination degree values, respectively, and the distance between the upper and lower curves represents the specific value. Overall, the coupling degree, coordination degree, and coupling coordination degree all show an upward spiral trend. That is to say, although there is a rebound, the overall trend is improving. Specifically, the coupling degree is located at the innermost side, indicating that the coupling effect is the best among the three data sets, but the coupling degree fluctuates wildly. Compared with the coupling degree, the coordination degree is located on the outermost side. The coordination degree is the worst among the three data sets but has more minor fluctuations.

From the perspective of fluctuation trends, it can take three points, A, B, and C, as cut-off points and divide them into three stages: ①, ②, and ③. Then, it can be concluded that, unlike the ② and ③ intervals, in the ① interval, the coupling degree and the coordination degree are in the divergence direction. The coupling of rural development and governance is greatly affected by the T value of the coordination degree, and the coupling coordination degree of the two is poor. However, unlike intervals ② and ③, resource investment in rural development and governance during this period showed a double-low phenomenon, and the fluctuation range was not extensive. In the ② and ③ intervals, the coupling degree, coordination degree, and coupling coordination degree all show the same trend, offering a spiral evolution trend of wide opening and closing. This shows that rural development and governance at this stage are greatly affected by the internal and external environment, especially the policy environment, and offer large fluctuations. In addition, since 2012, the degree of coupling and coordination between development and governance has continued to move closer to high-quality coordination. This shows that the coupling and coordination relationship between rural development and governance is staged, fluctuating continuously in continuous, high-quality coordination. That is, before 2012 and after 2021, several coupling and coordination intersections will be formed.

Since 2000, at the national macro level, the No. 1 Central Document issued every year has focused on the issues of rural areas and farmers, pointing out the direction for rural development. In 2013, the strategy for modernizing the national governance system and governance capabilities was proposed. However, federal resources are more inclined to development content such as rural road network construction, rural industry development, and rural poverty alleviation. At the same time, the governance sector only focuses on partial content, such as improving human settlements. This is also the main reason for the trend divergence in the interval ①.

Implementing the targeted poverty alleviation policy in 2015 pushed rural development into the fast lane. Administrative resources and social capital continued to flow into the countryside, causing the coupling, coordination, and coupling coordination between development and governance to rise or fall simultaneously. With the emergence of the multiplier effect of administrative resources and social capital, the coupling coordination degree of development and governance gradually tended to high-quality coordination and reached the optimal state in 2018. The overall trend is as follows: interval ②.

Since 2018, with the implementation of the rural revitalization strategy, the previous urban-rural dual pattern has become the focus of contradiction. Each region has successfully established rural revitalization bureaus and set up urban and rural grassroots governance committees to jointly promote high-quality rural development and efficient governance. This is also the first time that rural development and governance have been put on the same footing on a factual level, and various policies and resources have been simultaneously directed towards the two dimensions of rural development and governance. This also makes the three curves highly synchronized in the interval ③.

## Significant positive correlation between the degree of coupling coordination and rural revitalization

The regression analysis results of the coupling coordination degree of rural development and governance and the comprehensive score of rural revitalization show that the coupling coordination degree of rural development and governance can explain 82.1% of the rural revitalization assessment value changes. During the F test, it was found that the model passed the F test (F = 36.780, p = 0.000<0.05), which means that the coupling coordination degree will impact the evaluation value of rural revitalization. The regression coefficient value of the coupling coordination degree is 0.129 (t = 6.065, p = 0.000<0.01), which means that the coupling coordination degree will have a significant positive impact on the rural revitalization evaluation value.

In summary, all coupling coordination degrees will significantly impact the value of the rural revitalization evaluation. To intuitively describe the positive relationship between the two, the following figure is drawn (Fig 4). It can be seen from the figure that the coupling coordination value of rural development and governance and the evaluation value of rural revitalization are consistent, and both show an increasing trend from left to right. From a mathematical logic point of view, the coupling coordination degree and the rural revitalization assessment value maintain a consistent slope; their upward trends are highly compatible. This further proves that the coupling of rural development and governance is closely related to the realization of rural revitalization, showing a trend of both increase and decrease.

Furthermore, during the period 2016–2018, the two trend lines intersected. Before the intersection, the rural revitalization assessment value's linear relationship is stronger than the coupling coordination degree's linear relationship. After the meeting, this situation is reversed, and the linear relationship of the coupling coordination degree was stronger than the linear relationship of the rural revitalization assessment value. This shows that the low degree of coupling coordination in the early stage was a limiting factor in realizing rural revitalization. After 2018, the improvement of coupling coordination has become a pulling factor for rural revitalization.

## Coupling and coordination path of development and governance to achieve rural revitalization

The coupling and coordination of rural development and governance is the key to solving the shortcomings and weaknesses of rural revitalization. As problems such as rural hollowing out

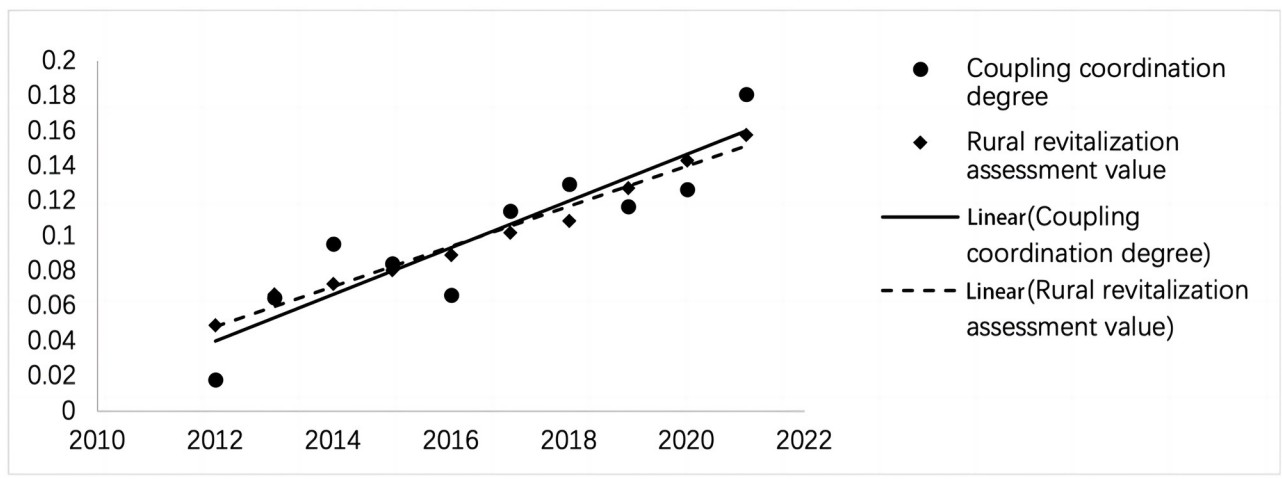

**Fig 4. Scatter distribution of coupling coordination degree and the rural revitalization evaluation value.**

and aging have intensified in recent years, rural development and governance have fallen into the dilemma of publicity, autonomy, tradition, and modernity, resulting in stagnant rural development and suspended rural governance. The root cause of this problem lies in the unclear cognitive logic of rural development and governance, inaccurate positioning of rural development and inefficient governance quality, insufficient mutual embedding of rural development and governance resources, and discrete and fragmented rural development and governance systems. Therefore, it can effectively respond to the urgent national need for rural revitalization to couple rural development and governance by exploring the coupling and coordination relationship between development and governance and optimizing the coupling path as the starting point.

Organizational isomorphism to build a coupling and coordination system for rural development and governance. Due to the weak development of the rural collective economy, the leadership ability of grassroots organizations has declined sharply, resulting in problems such as a single form of rural governance and weakened governance authority. Since 2016, many villages have begun to explore a rural collective economic development model with the two village committees as the leading team. However, this model promotes the overlap of rural administrative, economic development, and social governance rights, resulting in political and economic integration. The eastern region explored pilot reforms to separate politics and the economy to break this status quo. However, the applicability of the reform is not strong, and the separation of politics and economy cannot solve existing problems. Instead, it leads to the separation of economic development and social governance. Therefore, on the whole, the system of political and economic integration, through organizational isomorphism and governance embedding, can not only reconstruct the rural community of interests with the collective economy as the core but also reconstruct the organization and living community, making up for the severe brain drain in rural areas. A series of problems caused by villagers' low degree of the organization make rural governance more efficient.

Tilt resources and improve the supply of coupled and coordinated factors for rural development and governance. In terms of funding, the government should increase financial investment in rural development to ensure that rural areas receive sufficient development funds; financial institutions should develop financial products suitable for pastoral needs to improve the availability and sustainability of rural financial services; they should also Create a good investment environment, encourage the private sector to invest in rural projects, and attract private funds. From the technical dimension, the government and agricultural research institutions should strengthen agricultural technology research and development and provide advanced agrarian production technologies and management methods to improve the quality and yield of farm products; promote digital transformation in rural areas, provide Internet access and e-commerce support, and encourage rural e-commerce and the development of online sales of farm products. In terms of talent, the government should increase investment in rural education, improve the quality of education, and ensure that rural students have access to educational resources equivalent to urban students; encourage and attract urban talents and professionals to return to rural areas to work or start businesses, and inject new energy into rural development. Strengthen the training of rural leading cadres, improve their management and decision-making capabilities, and promote the modernization of rural governance. Regarding the labor force, the government should encourage the diversification of rural industries, create more job opportunities, and reduce the outflow of rural labor. Establish a sound rural social security system, including rural pensions, medical care, and unemployment insurance, to improve rural labor's quality of life and sense of security. Provide skills training for the farmers to make adapting to emerging industries and employment opportunities easier.

Operating mechanism to optimize rural development and governance's coupling and coordination path. First, institutional embeddedness is the basis of rural governance, including laws, rules, policies, etc., which play a crucial role in rural development. A sound institutional framework must be established to promote sustainable rural economic, social, and environmental development to improve rural development and governance. Second, resource embedding is the driving force for rural development, and the rational allocation and management of resources will help improve the efficiency and sustainability of rural development. The government should provide sufficient development funds to guide investment into rural industries and infrastructure construction. Third, technology embedding is the key to rural development. Modern technology can improve the productivity and competitiveness of rural industries and promote the transformation and upgrading of the rural economy. In short, improving the operating mechanism of rural development and governance requires taking measures in the three dimensions of institutional embedding, resource embedding, and technology embedding. By establishing a sound institutional framework, enriching resource support, and promoting scientific and technological innovation, we can promote sustainable rural development, improve the living standards of rural residents, and successfully implement the rural revitalization strategy.

## Conclusion

High-quality development and high-efficiency governance are important ways to achieve Chinese-style modernization. Rural areas are an essential part of grassroots governance, and grassroots modernization is the cornerstone of the country's comprehensive modernization. Based on this, rural areas need both development and governance. How to handle the relationship between development and governance has become the key to coordinating sequential conflicts and stimulating the vitality of rural revitalization. Based on the data from the China Rural Statistical Yearbook, China Urban and Rural Statistical Yearbook, and China Civil Affairs Statistical Yearbook, the study selected and extracted the evaluation index set to construct the rural development and governance evaluation and the rural revitalization evaluation index system. Based on the evaluation system, the entropy method, coupling coordination degree model, and regression analysis model are combined to evaluate the coupling coordination relationship between rural development and governance, and the relationship between coupling coordination and rural revitalization results, and then answer the three significant hypotheses proposed in the study.

Through evaluation, the study found that rural development and governance present a spiral coupling and coordination relationship. That is to say, although there are back-and-forth fluctuations, the overall trend is improving. The degree of coupling coordination has a significant positive correlation with rural revitalization. As the degree of coupling coordination increases, the effect of rural revival gradually increases, and the two have a strong positive relationship. Based on the two conclusions above, the study believes that the coupling path of development and governance of rural revitalization can be discussed in three dimensions. The first is an organizational isomorphism, which builds a coupled coordination system for rural development and governance. The second is to tilt resources and improve the supply of connected and coordinated factors for rural development and governance. The third is the operating mechanism to optimize rural development and governance's coupling and coordination path. This research result is a breakthrough in past rural research. Rather than focusing solely on rural development or governance, the study emphasizes both development and governance. It also uses data to demonstrate the critical role of the coupling of rural development and governance in rural revitalization. In theory, it introduces a new research perspective to the

countryside, and in practice, it provides targeted suggestions for the scheduled realization of rural revitalization.

## Supporting information

**S1 Data.**
(XLSX)

## Acknowledgments

The data used in this study come from statistical yearbooks published by the government. The analysis in this article uses SPSS 26.0. We are grateful to local government websites for sharing statistics and for the technical support provided by SPSS software.

## Author Contributions

**Conceptualization:** Hongxun Xiang, Boleng Zhai.

**Data curation:** Hongxun Xiang, Boleng Zhai, Yang Yang.

**Formal analysis:** Hongxun Xiang, Yang Yang.

**Investigation:** Hongxun Xiang, Boleng Zhai, Yang Yang.

**Methodology:** Hongxun Xiang, Boleng Zhai, Yang Yang.

**Project administration:** Hongxun Xiang.

**Resources:** Hongxun Xiang.

**Software:** Hongxun Xiang.

**Supervision:** Hongxun Xiang, Yang Yang.

**Validation:** Hongxun Xiang, Boleng Zhai, Yang Yang.

**Visualization:** Hongxun Xiang, Boleng Zhai.

**Writing – original draft:** Hongxun Xiang.

**Writing – review & editing:** Boleng Zhai, Yang Yang.

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
