## [Decision Letter · Decision Letter 0]

23 Jan 2024

PONE-D-23-41913The realization logic of rural revitalization: Coupled coordination analysis of development and governance.PLOS ONE

Dear Dr. xiang,

Thank you for submitting your manuscript to PLOS ONE. After careful consideration, we feel that it has merit but does not fully meet PLOS ONE’s publication criteria as it currently stands. Therefore, we invite you to submit a revised version of the manuscript that addresses the points raised during the review process.

We look forward to receiving your revised manuscript.

Kind regards,

Liang Zhuang, Ph.D.

Academic Editor

PLOS ONE

A clean copy of the edited manuscript (uploaded as the new *manuscript* file)”.

Reviewers' comments:

Reviewer's Responses to Questions

**Comments to the Author**

1. Is the manuscript technically sound, and do the data support the conclusions?

Reviewer #1: Yes

Reviewer #2: Yes

2. Has the statistical analysis been performed appropriately and rigorously? 

Reviewer #1: Yes

Reviewer #2: Yes

3. Have the authors made all data underlying the findings in their manuscript fully available?

Reviewer #1: Yes

Reviewer #2: Yes

4. Is the manuscript presented in an intelligible fashion and written in standard English?

Reviewer #1: Yes

Reviewer #2: Yes

5. Review Comments to the Author

Reviewer #1: The study found that current rural development and governance present a spiral coupling coordination relationship, and the degree of coupling coordination significantly correlates with rural revitalization. It proposed three development and governance coupling and coordination paths for rural revival: organizational isomorphism, resource tilting, and mechanism guarantee. However, The language, structure, and some details of the article need to be further modified to meet the requirements of publication. The specific opinions are as follows:

1. The abstract is too lengthy. Please briefly describe the background, clarify the scientific problem, and the results need to be concise and concise. The conclusion needs to be further summarized and condensed.

2. L32-34, please check the language gramma of the two sentences.

3. L130, please modify punctuation marks.

4. L132, please check the sentence.

5. The language of the entire article needs further polishing and improvement.

6. Literature review: the author is only listing the research of previous scholars. Please highlight the main idea of each paragraph and enhance the logical coherence in order to better introduce the scientific issues of the article.

7. Materials and methods: consider merging the description of this section into the preface and adding descriptions of indicators in Tables 1 and 2, such as calculation formulas and measurement of qualitative indicators.

8. Avoid using the first person in the whole article.

9. L310-311, Is this a reference to previous research? Please add.

10. Data Sources: this section should be placed in “Materials and methods”.

11. Discussion: the discussion is too lengthy and lacks comparison with others, requiring further improvement.

12. Suggest placing the limitations of the conclusion section on this article in the discussion section, and highlight the innovative results of this study in the conclusion.

Reviewer #2: Rural development and governance are critical to rural revitalization. Exploring the relationship between rural development and governance and understanding the impact of this relationship on rural revival are key to addressing current rural challenges and achieving integrated development. The article employs a regression analysis model to investigate the coupling results of rural development and governance and their relationship with rural revitalization. This analysis holds positive implications for gaining insights into rural revitalization at the current stage.

Revisions are as follows:

1. Minimize the use of interrogative sentence structures in the introduction and prefer declarative sentences. Enrich the explanation of the research background, and emphasize the theoretical and practical significance of this study throughout the article.

2. Provide specific points and descriptions in the conclusion and discussion sections.

3. The literature review should not merely list authors and their works but should elaborate on their viewpoints and significant contributions to the field. Address shortcomings in past studies and highlight the innovations and advantages of the current research in comparison. Additionally, discuss future research directions and prospects.

6. PLOS authors have the option to publish the peer review history of their article (what does this mean?). If published, this will include your full peer review and any attached files.

Reviewer #1: No

Reviewer #2: No

---

## [Author Response · Author response to Decision Letter 0]

21 Feb 2024

Response to Reviewers 1

[Comments 1] The abstract is too lengthy. Please briefly describe the background, clarify the scientific problem, and the results need to be concise and concise. The conclusion needs to be further summarized and condensed.

Response: Thank you for your valuable comments to us. We rewrote the abstract, especially the background(Pg1, Ln12-33), methods(Pg1, Ln35-45), and results(Pg2, Ln47-68), to make it more precise. Specific changes can be found in a marked copy of the manuscript. 

[Comments 2] L32-34, please check the language gramma of the two sentences.

Response: Thank you for your valuable comments to us. We rewrote the entire paragraph, using shorter sentences instead of longer ones to make it more straightforward(Pg4, Ln143-153). Specific changes can be found in a marked copy of the manuscript. 

[Comments 3 & 4 & 6] L130, please modify punctuation marks. & L132, please check the sentence. & Literature review: the author is only listing the research of previous scholars. Please highlight the main idea of each paragraph and enhance the logical coherence in order to better introduce the scientific issues of the article.

Response: Thank you for your valuable comments to us. We have put together responses to these questions because we have rewritten the content of the literature review. Compared with the previous literature review, we made the following improvements. First, we changed the literature review format to make each paragraph's main idea more prominent. For example, we divided research on rural development into four categories and research on rural governance into three categories and summarized each. Secondly, we reorganized the literature context according to the logic from shallow to deep, making the logic of the article review more coherent. Third, we summarize the shortcomings of past research and future development trends to lay a theoretical foundation for the study. Specific changes can be found in a marked copy of the manuscript(Pg5, Ln189-308). 

[Comments 5 & 8] The language of the entire article needs further polishing and improvement. 

Response: Thank you for your valuable comments to us. We have re-examined the language issues of the full text and made improvements mainly in the following aspects. The first is to change long sentences into short sentences, such as P4 and Ln148-153. Secondly, we modify ten first-person problems, such as P4 and Ln162-168. Finally, we specifically checked the use of articles "the," "a," and "an" and corrected 14 grammatical errors. We have also revised some other grammatical issues. Specific changes can be found in a marked copy of the manuscript.

[Comments 7] Materials and methods: consider merging the description of this section into the preface and adding descriptions of indicators in Tables 1 and 2, such as calculation formulas and measurement of qualitative indicators.

Response: Thank you for your valuable comments to us. We have added a description of the problem in the introduction. The description of the research framework in the Materials and Methods section is used to respond to the questions raised in the introduction. Still, the content of the description has been modified. The issues already discussed in the preface will not be discussed here. In addition, descriptions of some general problems have been deleted from this section and added to the preface. The revised article has a more reasonable structure(P8, Ln339-399). Regarding the indicator description, due to the impact of page length, it is not easy to add content to Tables 1 and 2. Therefore, we put the description and calculation process of the indicators into the main text(P12, Ln478-507). 

[Comments 9] L310-311, Is this a reference to previous research? Please add.

Response: Thank you for your valuable comments to us. This result references previous results, and we have added a piece of literature here. This document is the result of our previous research and is used to support the conclusions in the article. [Heng X, Xiang HX. Consistency or Conflict: Policy Reason and Value Choice in Urban and Rural Community Governance. Administrative Tribune. 2023 Jan;30(01):118-125.]

[Comments 10] Data Sources: this section should be placed in “Materials and methods”.

Response: Thank you for your valuable comments to us. We have restructured and removed data sources from the results and added them to Materials and Methods(P14, Ln521-540).

[Comments 11] Discussion: the discussion is too lengthy and lacks comparison with others, requiring further improvement.

Response: Thank you for your valuable comments to us. We have re-adjusted the content of the discussion section and mainly made the following changes. First, the content of the discussion section has been streamlined to make it more focused. Secondly, a comparative analysis with related research results has been added to the discussion section, and valuable parts of other results have been borrowed. Third, we modified the structure of the discussion section to make it easier to read. Specific changes can be found in a marked copy of the manuscript(P17, Ln647-1048).

[Comments 12] Suggest placing the limitations of the conclusion section on this article in the discussion section, and highlight the innovative results of this study in the conclusion.

Response: Thank you for your valuable comments to us. We have removed the discussion of limitations in the Conclusion and added it to the main body of the Discussion. In addition, we modified the conclusion's content to highlight this study's innovative results. 

Response to Reviewers 2

[Comments 1] Minimize the use of interrogative sentence structures in the introduction and prefer declarative sentences. Enrich the explanation of the research background, and emphasize the theoretical and practical significance of this study throughout the article. 

Response: Thank you for your valuable comments to us. We deleted the introduction's interrogative expression and changed it to a declarative sentence. In addition, we rewrote parts of the introduction and added a description of theoretical and practical implications(P5, Ln187-202).

[Comments 2] Provide specific points and descriptions in the conclusion and discussion sections. 

Response: Thank you for your valuable comments to us. We have rewritten the Conclusion and Discussion sections. In the discussion section, we have made the following optimizations: First, we have streamlined the content to make it more specific. Secondly, a comparative analysis with related research results is added to the discussion section to highlight the uniqueness of this study. Third, we modified the structure of the discussion section to make it easier to read. In the Conclusion section, we have removed the discussion of limitations and added it to the main body of the Discussion. In addition, we modified the conclusion's content to highlight this study's innovative results(P17, Ln647-1108).

[Comments 3] The literature review should not merely list authors and their works but should elaborate on their viewpoints and significant contributions to the field. Address shortcomings in past studies and highlight the innovations and advantages of the current research in comparison.

Response: Thank you for your valuable comments to us. Compared with the previous literature review, we made the following improvements. First, we changed the literature review format to make each paragraph's main idea more prominent. For example, we divided research on rural development into four categories and research on rural governance into three categories and summarized each. Secondly, we reorganized the literature context according to the logic from shallow to deep, making the logic of the article review more coherent. Third, we summarize the shortcomings of past research and future development trends to lay a theoretical foundation for the study. Specific changes can be found in a marked copy of the manuscript(Pg5, Ln189-308).

---

## [Decision Letter · Decision Letter 1]

30 Apr 2024

PONE-D-23-41913R1The realization logic of rural revitalization: Coupled coordination analysis of development and governance.PLOS ONE

Dear Dr. xiang,

Thank you for submitting your manuscript to PLOS ONE. After careful consideration, we feel that it has merit but does not fully meet PLOS ONE’s publication criteria as it currently stands. Therefore, we invite you to submit a revised version of the manuscript that addresses the points raised during the review process.

We look forward to receiving your revised manuscript.

Kind regards,

Liang Zhuang, Ph.D.

Academic Editor

PLOS ONE

Journal Requirements:

Reviewers' comments:

Reviewer's Responses to Questions

**Comments to the Author**

1. If the authors have adequately addressed your comments raised in a previous round of review and you feel that this manuscript is now acceptable for publication, you may indicate that here to bypass the “Comments to the Author” section, enter your conflict of interest statement in the “Confidential to Editor” section, and submit your "Accept" recommendation.

Reviewer #1: All comments have been addressed

2. Is the manuscript technically sound, and do the data support the conclusions?

Reviewer #1: Yes

3. Has the statistical analysis been performed appropriately and rigorously? 

Reviewer #1: Yes

4. Have the authors made all data underlying the findings in their manuscript fully available?

Reviewer #1: Yes

5. Is the manuscript presented in an intelligible fashion and written in standard English?

Reviewer #1: Yes

6. Review Comments to the Author

Reviewer #1: The authors have revised most of my comments, but I still have two questions as follows:

Firstly, the language of the whole text needs further refinement.

Secondly, the conclusion section, especially the recommendations, in the abstract should be specific. For example, what are the three coupling coordination paths?

7. PLOS authors have the option to publish the peer review history of their article (what does this mean?). If published, this will include your full peer review and any attached files.

Reviewer #1: No

---

## [Author Response · Author response to Decision Letter 1]

6 May 2024

Response to Reviewers 1

We greatly appreciate your help in further improving our research. Your valuable suggestions allow us to improve our research results further. Below are responses to your comments. Thank you again for your help.

[Comments 1] Firstly, the language of the whole text needs further refinement.

Response: Thank you for your valuable comments. We checked the language issues of the full text through self-checking and help from others. We have modified the language expression of all contents, including abstract, main text, acknowledgments, etc. Pay special attention to issues such as articles, tenses, and persons. At the same time, we modified the long sentences of the article to express its meaning better. Finally, we reconfirmed the expression of some proper nouns to ensure their correctness. We have retained traces of modifications in the manuscript for your review. 

[Comments 2] Secondly, the conclusion section, especially the recommendations, in the abstract should be specific. For example, what are the three coupling coordination paths?

Response: Thank you for your valuable comments. There is indeed a problem with our previous expression. We have made corrections to this version. We have added detailed research conclusions, countermeasures, and suggestions to make the abstract more specific (Pg2, Ln47-54). 

Thank you again for your help with our research.

---

## [Decision Letter · Decision Letter 2]

2 Jun 2024

The realization logic of rural revitalization: Coupled coordination analysis of development and governance.

PONE-D-23-41913R2

Dear Dr. xiang,

We’re pleased to inform you that your manuscript has been judged scientifically suitable for publication and will be formally accepted for publication once it meets all outstanding technical requirements.

Kind regards,

Liang Zhuang, Ph.D.

Academic Editor

PLOS ONE

Additional Editor Comments (optional):

Reviewers' comments:

Reviewer's Responses to Questions

**Comments to the Author**

1. If the authors have adequately addressed your comments raised in a previous round of review and you feel that this manuscript is now acceptable for publication, you may indicate that here to bypass the “Comments to the Author” section, enter your conflict of interest statement in the “Confidential to Editor” section, and submit your "Accept" recommendation.

Reviewer #1: All comments have been addressed

2. Is the manuscript technically sound, and do the data support the conclusions?

Reviewer #1: Yes

3. Has the statistical analysis been performed appropriately and rigorously? 

Reviewer #1: Yes

4. Have the authors made all data underlying the findings in their manuscript fully available?

Reviewer #1: Yes

5. Is the manuscript presented in an intelligible fashion and written in standard English?

Reviewer #1: Yes

6. Review Comments to the Author

Reviewer #1: The author has responded well to all the previous suggestions and I recommend accepting this manuscript for publication.

7. PLOS authors have the option to publish the peer review history of their article (what does this mean?). If published, this will include your full peer review and any attached files.

Reviewer #1: No

---

## [Editor Report · Acceptance letter]

14 Jun 2024

PONE-D-23-41913R2 

PLOS ONE

Dear Dr. xiang, 

I'm pleased to inform you that your manuscript has been deemed suitable for publication in PLOS ONE. Congratulations! Your manuscript is now being handed over to our production team.

Kind regards, 

on behalf of

Professor Liang Zhuang 

Academic Editor

PLOS ONE